# Development of rechargeable high-energy hybrid zinc-iodine aqueous batteries exploiting reversible chlorine-based redox reaction

Guojin Liang[1], Bochun Liang[1], Ao Chen[1], Jiaxiong Zhu[1], Qing Li[1], Zhaodong Huang[1], Xinliang Li [1], Ying Wang [2] ✉, Xiaoqi Wang [3], Bo Xiong[3], Xu Jin[3], Shengchi Bai[3], Jun Fan [1] ✉ & Chunyi Zhi [1,4] ✉

The chlorine-based redox reaction (ClRR) could be exploited to produce secondary high-energy aqueous batteries. However, efficient and reversible ClRR is challenging, and it is affected by parasitic reactions such as $Cl_2$ gas evolution and electrolyte decomposition. Here, to circumvent these issues, we use iodine as positive electrode active material in a battery system comprising a Zn metal negative electrode and a concentrated (e.g., 30 molal) $ZnCl_2$ aqueous electrolyte solution. During cell discharge, the iodine at the positive electrode interacts with the chloride ions from the electrolyte to enable interhalogen coordinating chemistry and forming $ICl_3^-$. In this way, the redox-active halogen atoms allow a reversible three-electrons transfer reaction which, at the lab-scale cell level, translates into an initial specific discharge capacity of 612.5 mAh $g_{I2}^{-1}$ at 0.5 A $g_{I2}^{-1}$ and 25 °C (corresponding to a calculated specific energy of 905 Wh $kg_{I2}^{-1}$). We also report the assembly and testing of a Zn‖Cl-I pouch cell prototype demonstrating a discharge capacity retention of about 74% after 300 cycles at 200 mA and 25 °C (final discharge capacity of about 92 mAh).

The chlorine ($Cl_2$) redox reaction is utilized for gas–liquid phase-conversion reactions and exhibits a reaction potential of 1.36 V versus SHE (standard hydrogen electrode at 25 °C) and a theoretical gravimetric capacity of 756 mAh $g^{-1}$ based on a one-electron Cl-based conversion reaction[1,2]. Looking into the ever efforts on developing $Cl_2$-based batteries, the earliest prototype was first proposed in 1884 as a $Zn‖Cl_2$ battery system and was further utilized with a 500 kW h battery run by Energy Development Associates in the 1980s[3]. However, research on Cl-redox electrodes was largely limited within the past four decades due to the leakage of gaseous $Cl_2$, causing safety

concerns. Although the Cl-redox feature was recently exploited by confining the oxidized $Cl^{-0.25}$ inside a graphite interlayer based on intercalation chemistry[4], the most common approach to fix the oxidized $Cl_2$-based cathode is to apply adsorption-type host materials, such as activated carbon (AC) and graphite[5–8], in which interactions between $Cl_2$ and the carbon-based host materials involve physical adsorption. As a result, the diffusion of gaseous toxic $Cl_2$ could not be fully eliminated. Another possible strategy is to store gaseous $Cl_2$ in low-temperature water (below 9.6 °C) to form a solid-state chlorine hydrate as $Cl_2 \cdot x H_2O$ ($x \approx 5.9$)[9,10], in which the electrochemically active

[1]Department of Materials Science and Engineering, City University of Hong Kong, Kowloon, China. [2]State Key Laboratory of Rare Earth Utilization, Changchun Institute of Applied Chemistry, Chinese Academy of Sciences, Changchun, China. [3]Research Institute of Petroleum Exploration & Development (RIPED), Research Center of New Energy, Beijing, PR China. [4]Center for Advanced Nuclear Safety and Sustainable Development, City University of Hong Kong, Kowloon, Hong Kong, China. ✉e-mail: ywang_2012@ciac.ac.cn; junfan@cityu.edu.hk; cy.zhi@cityu.edu.hk

$Cl_2$ gas is released by heating once the Zn‖$Cl_2$ battery is discharged. However, the main drawback is the necessity of a complex thermal and gas management system to run the gaseous $Cl_2$ electrode loop[9,10].

Irreversible Cl-redox reactions in battery electrodes generally occur in aqueous electrolytes not only because of the $Cl_2$ fixing issue at the positive electrode side but also because of the competing oxygen evolution reaction (OER as 1.23 V versus SHE at 25 °C), which exhibits an onset potential 0.13 V lower than that needed to trigger the $Cl_2$ evolution reaction (ClER) (similar to the competing relationship between the OER and ClER in the seawater splitting process[11]). In addition, the tendency of $Cl_2$ to solubilize in water aggravates the instability of the $Cl_2$ electrode[12]. To avoid the OER, the aqueous electrolytes were replaced with molten metal chlorides that operate at high temperature (for example >130 °C in ref. 7); however, the flowing of a gaseous $Cl_2$ electrode was still necessary in the Li‖$Cl_2$ and Al‖$Cl_2$ cell configurations[7,13,14]. Regarding the metal‖$Cl_2$ configurations, Na‖$Cl_2$ and Li‖$Cl_2$ batteries have been recently developed by confining the oxidized $Cl_2$ inside highly microporous carbon[15]. However, stabilizing reversible Cl-based redox reactions (ClRRs) in aqueous electrolytes is challenging because of the drawbacks associated with the electrodes and the electrolyte, i.e., fixing the oxidized $Cl^0$ without producing gaseous $Cl_2$ and simultaneously ensuring that the oxidation reaction potential of $Cl^-$ is below the OER.

Based on this state-of-the-art knowledge, we further explore feasible strategies to exploit ClRR for battery applications. The selection principles are utilized to fix agents at the electrodes and electrolyte. Regarding fixing agents for oxidized $Cl^0$ at the electrode side, the ideal goal is to store $Cl^0$ in the solid state; the strategy of fixing $Cl_2$ by $H_2O$ molecules in solid-state chlorine hydrate ($Cl_2 \cdot xH_2O$)[1,16] sheds light on the feasibility of exploiting other coordinating agents to fix oxidized $Cl^0$. Theoretically, there is one basic requirement for the coordinating agents, namely, they should form strong chemical bonds with oxidized $Cl^0$. Given the Lewis acid nature of $Cl^0$, i.e., its strong electrophilicity, $Cl^0$ spontaneously produces gaseous $Cl_2$ molecules in the energy stable form; thus, it should be more energetically favourable for coordinating agents to bind with $Cl^0$ and prevent the production of $Cl_2$ molecules. Based on this consideration, we examined I atoms with high electronegativity and larger atomic weights[17], showing the possibility of using I atoms as fixing agents for $Cl^0$ to store the final products in terms of the solid state. On the other hand, regarding the electrolytes, the main goal is to regulate the $H_2O$ activity of the applied electrolytes to lower levels, retarding the onset potential of the OER and inhibiting the dissolving tendency of the final products of $Cl^0$.

In this research work, we apply I atoms as the coordinating agents to fix the oxidized $Cl^0$ based on interhalogen coordinating chemistry in a highly concentrated $ZnCl_2$ electrolyte, i.e., 30 m $ZnCl_2$ (where m is molality as mol kg$^{-1}$ calculated as moles of solute divided by the mass of the solvent) and a Zn metal negative electrode. Reversible ClRR can be obtained and deliver an average discharge voltage of 1.88 V and a discharge capacity of 210 mAh g$_{I2}^{-1}$ at 0.5 A g$_{I2}^{-1}$ and 25 °C (based on the mass of fixing agents of iodine unless otherwise specified, Supplementary Note 1). Two halogen-based redox centres based on Cl and I as the I-based positive electrode were examined, and the electrode accommodates cascade interhalogen reactions with three-electron transfer processes, i.e., two electrons from I and one electron from Cl. This interhalogen electrode in a hybrid Zn metal battery system enables a discharge capacity of 612.5 mAh g$_{I2}^{-1}$ at 0.5 A g$_{I2}^{-1}$ and 25 °C and an average discharge voltage of 1.48 V, which translates to a calculated specific energy of 905 Wh k g$_{I2}^{-1}$.

## Results and discussion

### Electrochemical investigation on the redox activity of iodine-based electrodes in Cl-ion aqueous electrolyte solutions

We first studied ClRR as an independent reaction. Therefore, the working potential range was confined within 1.7 V to 2 V to capture the electrochemical features of the ClRR. The electrolyte applied here was 30 m $ZnCl_2$ at 25 °C in a two-electrode pouch cell configuration. To verify the role of I in fixing ClRR, a blank control experiment was conducted in which the corresponding cyclic voltammetry (CV) and galvanostatic charge–discharge (GCD) profiles were collected from the I-containing and I-absent samples, respectively. There was one noticeable cathodic peak with the I-containing electrode, and the corresponding peak current density was 21 times larger than that of the I-absent electrode, i.e., 3.48 mA cm$^{-2}$ versus 0.165 mA cm$^{-2}$. Such pronounced contrast indicates the promotion of ClRR from I-containing electrodes (Fig. 1a).

The capacity of the cell with the I-absent electrode does not show an apparent discharging plateau from the Cl-redox reactions, which is assigned to the dominant surface charge storage capacity and slight $Cl^0$-redox capacity; together, these capacities contribute to the output capacity of the AC host in the positive electrode, as shown in Supplementary Fig. 1. On the other hand, the corresponding capacity of the I-containing electrode is 215 mAh g$_{I2}^{-1}$ at 0.5 A g$_{I2}^{-1}$ and 25 °C, and the charge storage mechanisms of the iodine (ClRR) and AC (capacitive) jointy contribute (Fig. 1b). The average discharging voltage is 1.88 V, which is different from the voltage plateau resulting from the iodine-based redox as $I^0 \leftrightarrow I^-$ (-1.25 V) and $I^{0+} \leftrightarrow I^-$ (-1.60 V). Notably, the valence state of iodine was maintained at +1 in the specified voltage window (1.7 V to 2 V), and the corresponding redox behaviours of the two I-based electron-transfer reactions transitioning from initial $I^-$ to $I^0$ and

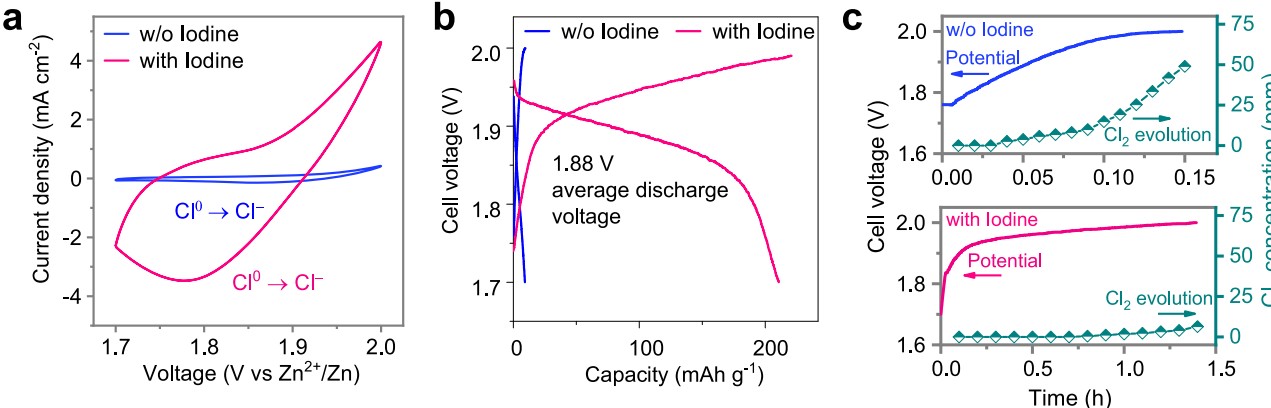

**Fig. 1 | Electrochemical characterizations of iodine activity towards ClRR. a** CV profiles at 1 mV s$^{-1}$ and **b** GCD profiles of the I-containing electrode cycled at 0.5 A g$_{I2}^{-1}$, and I-absent electrode cycled at 0.5 A g$_{AC}^{-1}$ based on AC mass, both in 30 m $ZnCl_2$ electrolytes and in two-electrode pouch cells at 25 °C. **c** *Operando* gaseous $Cl_2$ detection of $Cl_2$ evolution during the charging process of I-absent and I-containing electrodes. Note that the scales of the abscissas are different by 10-fold.

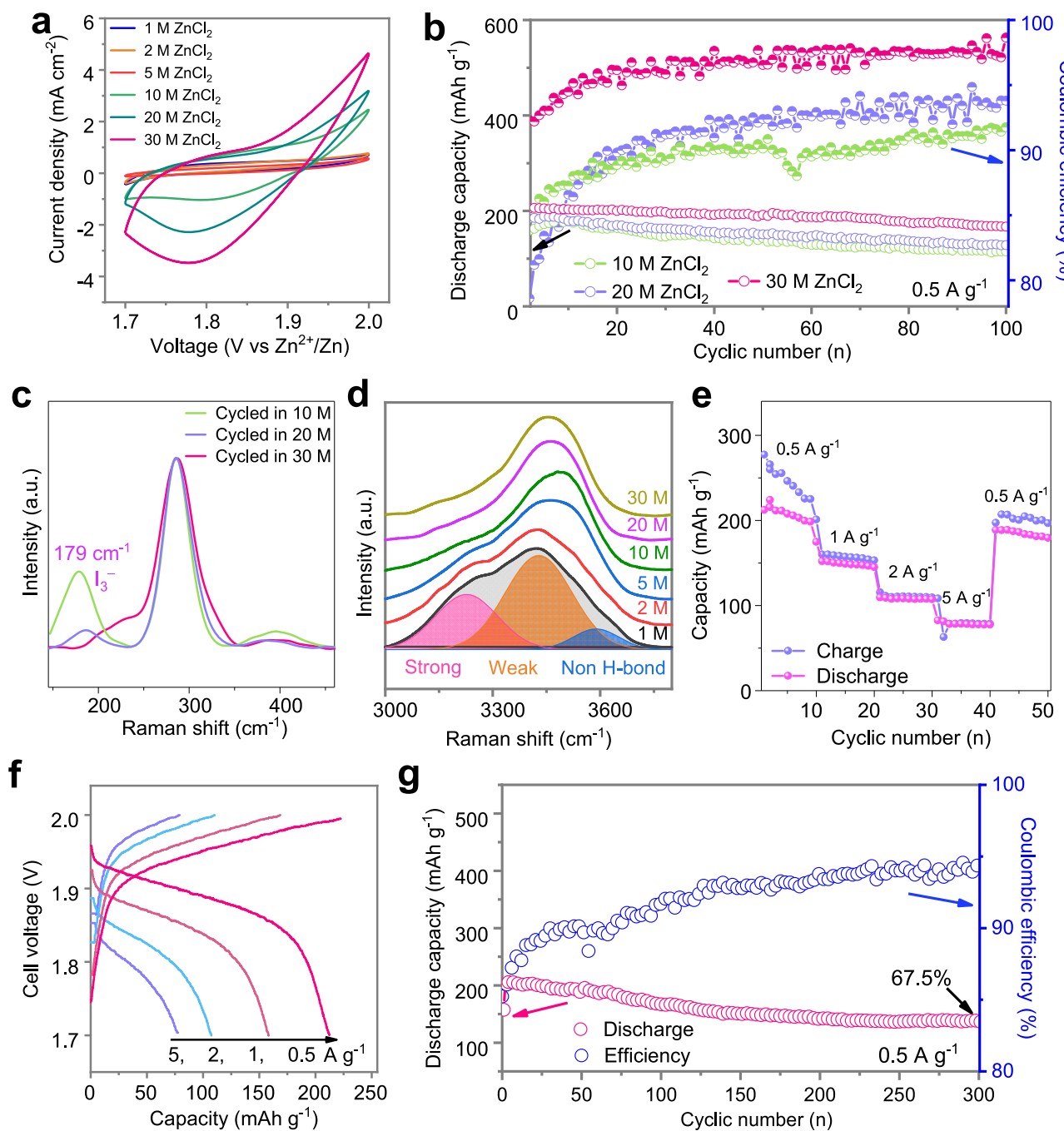

**Fig. 2 | Physicochemical and electrochemical characterizations of ZnCl₂-based aqueous electrolyte solutions at various concentrations. a** CV curves of the I-containing electrode in ZnCl₂ electrolytes with different concentrations at a scanning rate of 1 mV s⁻¹. **b** Cycling stability of ClRR in 10, 20, 30 m ZnCl₂ electrolytes at 0.5 A g$_{I2}$⁻¹, respectively, and the specific capacity was calculated based on the mass of I₂ in Zn‖Cl-I pouch cells at 25 °C. **c** Ex situ Raman profiles revealing the different hydrogen-bond networks of water in different electrolytes after 100 cycles corresponding to **b**. **d** Raman results of uncycled electrolytes with different concentrations, in which the deconvolution peaks of strong hydrogen bonds (H-bonds), weak H-bonds and non-H-bonds are centred at 3228 cm⁻¹, 3432 cm⁻¹, and 3594 cm⁻¹, respectively, for the 1 m ZnCl₂ electrolyte. **e** Rate performance of Zn‖Cl-I cells from 0.5 to 0.5 A g$_{I2}$⁻¹ and the corresponding GCD curves **f**. **g** Cycling stability of the Cl-redox reaction after 300 cycles at 0.5 A g$_{I2}$⁻¹.

to I⁺ are shown in Supplementary Fig. 2, in which GCD and in situ Raman measurements (in the 0.6 V to 1.75 V cell voltage range) and analyses are reported[18,19].

The cathodic peak in the CV curve shown in Fig. 1a correlates to one electron-transfer reaction, which can be assigned to the Cl⁰/Cl⁻ reduction reaction or extraction behaviour of the anionic ions as [ZnCl$_x$]²⁻ˣ in the carbon-based host at the positive electrode[20]. To confirm the mechanism, in situ measurements of the Cl2 gas were

carried out to monitor the Cl₂ evolution at both the positive and negative electrodes (Supplementary Fig. 3). For the I-containing electrode, no pronounced signals of Cl₂ gas were detected at different charge potentials from 1.7 V to 1.95 V, and the electrode released a small amount of Cl₂ from 1.95 V to 2.0 V (5.1 ppm at 2.0 V). In stark contrast, for the I-absent electrodes, gaseous Cl₂ was produced after charging to approximately 1.85 V and continuously increased during charging to 2.0 V (49.6 ppm at 2.0 V state) (Fig. 1c). The role of I as a

fixing agent for oxidized $Cl^0$ is schematically illustrated in Supplementary Fig. 4.

## Effect of the electrolyte concentration on the Cl-based redox reaction

After confirming the feasibility of applying I at the positive electrode of a Zn||AC cell to fix the ClRR, we further optimized electrolytes to support the cycling stability of the ClRR. As mentioned above, one main goal is to suppress the competing OER to obtain a highly reversible ClRR[21,22]. To optimize the electrolyte, linear sweep voltammetry (LSV) was measured for electrolytes with different $ZnCl_2$ concentrations from diluted 1 m to concentrated 30 m with the Ti-foil current collector as the positive electrode[23]; this was achieved by coupling with the Zn anode in a two-electrode cell configuration to identify the corresponding onset potentials of the OER and ClER (Supplementary Fig. 5a). With increasing electrolyte concentrations, the OER onset potential shifted to higher values, and conversely, the $Cl^-$ oxidation reaction shifted to lower working potentials (Supplementary Fig. 5b). The decrease in the $Cl^-$ oxidation potential can be explained by the Nernst equation associated with the increase in $Cl^-$ concentrations (Supplementary Note 2). In contrast to the 1 m $ZnCl_2$ electrolyte, the AC electrode tested in 30 m $ZnCl_2$ electrolyte produced a cathodic peak at 1.83 V in the CV profiles, further verifying the decrease in $Cl^-$ oxidation potential (Supplementary Fig. 5c). Therefore, the 30 m $ZnCl_2$ electrolyte is verified to decrease the $Cl^-$ oxidation potential below the OER potential, resulting in a suppressed OER.

We further studied the influence of electrolyte concentrations on the electrochemical behaviours of the I-containing electrode. Specifically, the corresponding CV curves in a two-electrode cell configuration show that the ClRR behaviour became relevant when the electrolyte concentration increased from 1 m to 30 m, i.e., 1, 2, 5, 10, 20, and 30 m (Fig. 2a). No reduction peaks are observed when the concentration is less than 10 m, which could be attributed to the competing OER that occurs before $Cl_2$ evolution. In contrast, reduction peaks appeared when the electrolyte concentrations increased to above 10 m, in which the ClRR dominated rather than the OER. This indicates that I as a coordinating agent can promote $Cl_2$ fixation well only in high-concentration electrolytes. On the other hand, the $b$-value for different scanning rates from 0.1 to 1 mV s$^{-1}$ approaches 0.5 of the reduction peak, corresponding to the nature of conversion reactions (detailed discussions on the $b$-value are provided in Supplementary Fig. 6).

We further studied the cycling performance of the ClRR in different electrolytes (10, 20, 30 m $ZnCl_2$) (Fig. 2b), in which the electrochemical stability was enhanced as the concentration increased. The low CE of the first cycle occurs due to the activation of $I_2$ in the positive electrode (Supplementary Fig. 7). Throughout 100 cycles for the Cl-redox reaction, the electrode capacity in the 30 m electrolyte remains at 167 mAh $g_{I2}^{-1}$, showing better performance compared to the 20 m case of 127 mAh $g_{I2}^{-1}$ and the 10 m case of 114 mAh $g_{I2}^{-1}$ (Fig. 2b). To trace the origin of these performance differences, cycled electrolytes were ex situ characterized via Raman measurements. The results showed that $I_3^-$ ions are present in lower concentration electrolytes (10 m, 20 m) but not in the 30 m case (Fig. 2c). This implies that the I species serving as $Cl^0$ coordinating agents dissolve in the low-concentration electrolytes and consequently result in losses of capacity during battery cycling. The activity of water determines the solubility of halide species, in which polar halide species, such as $I_3^-$ and $I^+$ ions, easily dissolve into polar $H_2O$ molecular networks according to the like-dissolves-like rule. Via analysis of the ex situ Raman measurements, it was verified that the deconvolution proportion of strong hydrogen bonds centred at 3228 cm$^{-1}$ decreases as the salt concentration increases; the tendency was increased for the nonhydrogen bonds (Fig. 2d and Supplementary Fig. 8)[24,25], indicating that the water activity is suppressed with reduced hydrogen bonds from 1 m to 30 m

electrolytes. Based on these analyses, 30 m electrolyte was selected as the suitable electrolyte for the ClRR.

To further explore the variations in different electrolyte systems, the solvation structures of the electrolytes, i.e., 1 m, 10 m, 20 m, and 30 m $ZnCl_2$ electrolytes, were analysed by molecular dynamics (MD) simulation. Specifically, snapshots of simulation boxes of various concentrations as well as zoomed-in images are displayed for the corresponding major solvation structure of $Zn^{2+}$ and their proportions in the system (Supplementary Fig. 9). In addition, the radial distribution functions (RDFs, g(r)) and coordination number (n(r)) of Zn-Cl and Zn-O were plotted to further determine the coordination structure of $Zn^{2+}$ ions (Supplementary Fig. 10). As $C_{ZnCl_2}$ increases, the coordination number of $H_2O$ decreases from approximately 5.6 to 1.6, while that of $Cl^-$ increases from approximately 0.4 to 3.8. Therefore, $Cl^-$ competes with $H_2O$ to coordinate $Zn^{2+}$ and gradually replaces the highly active coordinated $H_2O$ at high concentrations, which suppresses the reactivity of water and helps alleviate water-induced parasitic reactions. The ratio distributions of the corresponding solvation structures in various systems are calculated by setting the radius of the first solvation shell (i.e., the distance of the first valley in the RDF plot) as the cutoff distance.

At low concentrations, such as 1 m $ZnCl_2$ electrolyte, $Zn(H_2O)_6^{2+}$ is the major solvation structure, which takes up 64.44% $Zn^{2+}$ ions in the system, indicating the high reactivity of water towards $Zn^{2+}$. As the concentration increased, the $Cl^-$ ions gradually dominated the first solvation shell, with $Zn(H_2O)_3Cl_3^-$ and $Zn(H_2O)_4Cl_2$ becoming dominant at 10 m and 20 m. However, when the concentration increased to 30 m, the proportion of $Zn(H_2O)_3Cl_3^-$ and $Zn(H_2O)_4Cl_2$ decreased, while $Zn(H_2O)_2Cl_4^{2-}$ grew to occupy 32.07% of the $Zn^{2+}$ ions in the system (Supplementary Fig. 11). Notably, four-coordinated $ZnCl_4^{2-}$ occurs when the concentration increases to 20 m and becomes even more abundant at 30 m, which implies stronger binding between $Zn^2$ and $Cl^-$.

The binding energy of one water molecule with the rest of the structure was calculated to explore the evolution of dominant solvation structures (Supplementary Fig. 12 and Supplementary Note 3). Specifically, the binding strength of one water molecule of $Zn(H_2O)_6^{2+}$ in 1 m $ZnCl_2$ of −34.77 kcal mol$^{-1}$ is much higher than that for $Zn(H_2O)_4Cl_2$ and $Zn(H_2O)_2Cl_4^{2-}$ in high concentration electrolytes (−3.09 kcal mol$^{-1}$ and −2.77 kcal mol$^{-1}$, respectively), indicating that water molecules remain much more stable in $Zn(H_2O)_6^{2+}$. The low water binding energy of $Zn(H_2O)_4Cl_2$ and $Zn(H_2O)_2Cl_4^{2-}$ and the small difference of 0.32 kcal mol$^{-1}$ between them imply that water molecules in these structures are easily pulled out and replaced by $Cl^-$ ions in highly concentrated electrolytes. On the other hand, the reactivity of water can also be suppressed by breaking the hydrogen bond (H-bond) network in the aqueous electrolyte as $C_{ZnCl_2}$ (the concentration) increases. The H-bonds in different simulation boxes are shown to explore the impact of $C_{ZnCl_2}$ on the H-bonds formed between water and water (Supplementary Fig. 13), from which we can intuitively observe the decline of the H-bonds number from low concentration to high concentration. Specifically, we calculated the average hydrogen bond number formed by one water molecule for each concentration (Supplementary Fig. 14). The average H-bond number of one water molecule is 1.21 in 1 m $ZnCl_2$, while the H-bond number decreases to 0.53, 017, and 0.08 in the 10 m, 20 m, and 30 m systems, respectively. These results demonstrate that the H-bond network is broken as the concentration of $ZnCl_2$ ($C_{ZnCl_2}$) increases, which also contributes to the lower reactivity of water in the high $C_{ZnCl_2}$ case. This is consistent with the electrochemical stability window test in which the OER was suppressed to enable the stable realization of the $Cl^-$ ion redox reaction.

Regarding the electrochemical behaviour of the ClRR, the rate performance and corresponding GCD profiles were characterized at various operating specific currents in a Zn||Cl-I pouch cell at 25 °C (Fig. 2e, f). The ClRR enables a cell discharge capacity of 210 mAh $g_{I2}^{-1}$

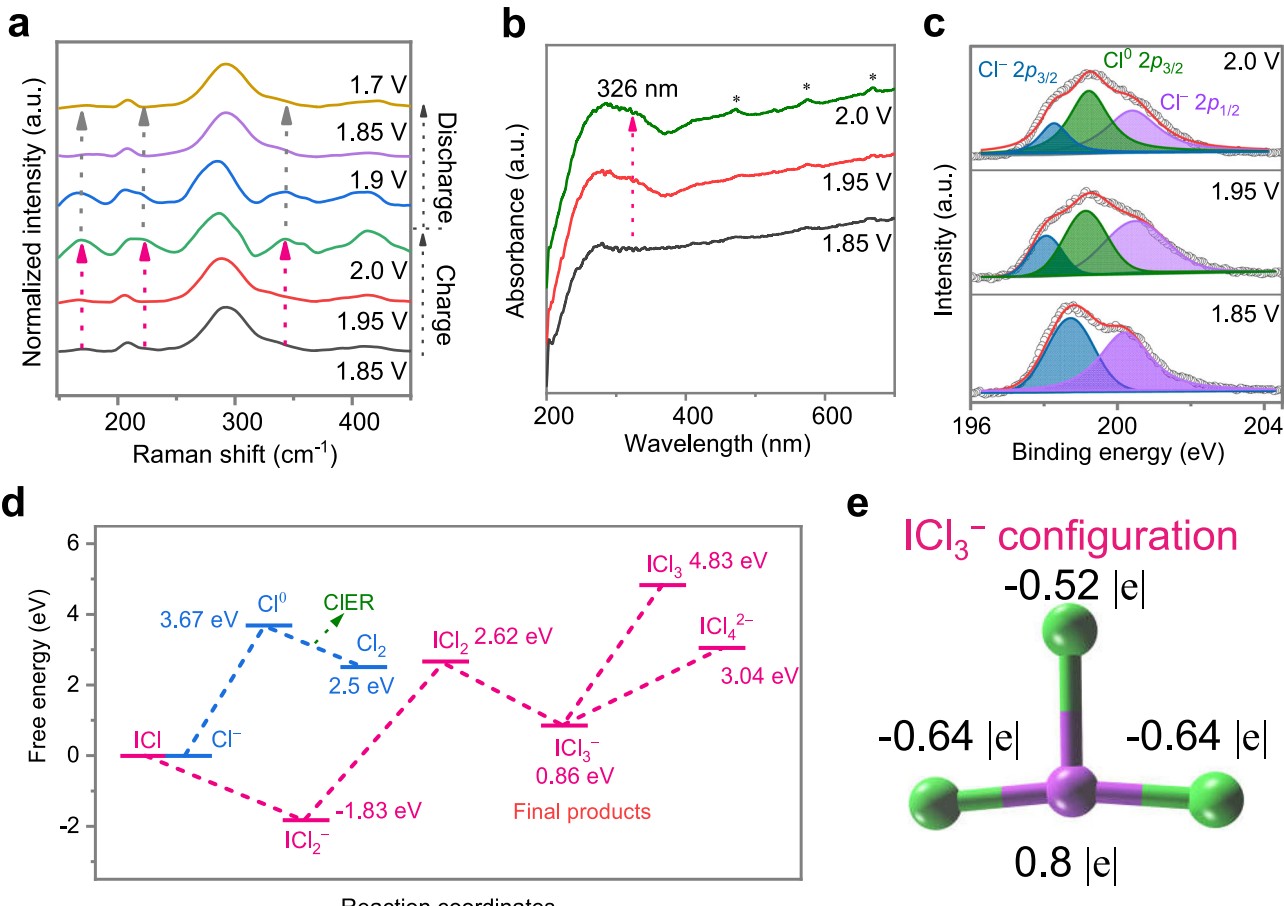

**Fig. 3 | Spectroscopic and computational characterizations of CIRR. a** In situ Raman characterizations of different charged and discharged states of the Cl-redox reaction. **b** Ex situ UV–vis results and **c** ex situ XPS to detect the Cl-I electrode at different charged states. **d** Energy profiles of possible reaction pathways with different corresponding products. **e** Configurational structure and valence states (atomic Mullikan charge) of $ICl_3^-$ as the final product.

with an average cell discharge voltage of 1.88 V at 0.5 A g$^{-1}$ for the 20$^{th}$ cycle, correlating to a specific energy of 388.5 Wh k g$_{I2}^{-1}$. In addition, it could retain 75 mAh g$_{I2}^{-1}$ at a specific current of 5 A g$^{-1}$, delivering a specific power of 1725 W k g$_{I2}^{-1}$. Long cycling stability was achieved with a discharge capacity of 140.8 mAh g$_{I2}^{-1}$ and 67.5% retention after 300 cycles at 0.5 A g$_{I2}^{-1}$ (Fig. 2g).

### Physicochemical characterizations of the I-containing positive electrode before and after cycling

A set of spectroscopy experiments and calculations were performed to depict the interacting manners and structural evolution of I-containing positive electrodes during Cl oxidation/reduction. First, in situ Raman spectroscopy was performed to measure five charge−discharge states marked in the GCD curves (Fig. 3a and Supplementary Fig. 15). Two Raman peaks are always present at different charged/discharged states. Specifically, the first peak centred at 204 cm$^{-1}$ is correlated to the presence of I$^+$Cl$^-$ (Supplementary Fig. 2), while the peak at 291 cm$^{-1}$ originates from the Cl$^-$ coordinating Zn$^{2+}$ species adsorbed on the electrode as hydrated [ZnCl$^{2+x}$(H$_2$O)$_y$]$^{x-}$ in the electrolyte. During the charging process, three Raman peaks appeared at 172 cm$^{-1}$, 223 cm$^{-1}$, and 345 cm$^{-1}$, which disappeared during the reverse discharging process. The newly emerging rather than shifting Raman peaks indicate that new species are produced, which correlates to the presence of new structural configurations after Cl$^0$ oxidation[26].

The ex situ ultraviolet/visible (UV/vis) spectroscopy of different charged states showed that the peak at 204 nm remained stationary as the peak intensity increased, and the peak at 326 nm originating from

Cl-I bonding intensified; this might occur due to the oxidation from Cl$^-$ to Cl$^0$ (Fig. 3b) since Cl$^0$ was previously verified at a peak position of approximately 330 nm[27,28]. The new peaks at 472 nm, 573 nm and 669 nm might be attributed to the new configuration of the final products. Furthermore, ex situ X-ray photoelectron spectroscopy (XPS) was performed to analyse the valence states of Cl in different charged states. The binding energy at 199 eV emerged, which corresponds to Cl$^0$ as the neutral valence state (Fig. 3c)[20,29]. This further verified the oxidation reaction from Cl$^-$ to Cl$^0$ to complete the conversion reaction. Of note, even though it was posttreated and analysed in a vacuum atmosphere for XPS measurement, Cl$^0$ was detected without any major evaporation, indicating that Cl was fixed by I. The XPS results of Cl at the initial and fully discharged states are shown in Supplementary Fig. 16. When checking the variations in the valence states of iodine as coordinating agents, the shape of the I 3$d$ peaks remained almost unchanged with a reversible 0.2 eV shift in binding energy in charged and discharged states (Supplementary Fig. 17). The slight shift in the valence states of I can be attributed to electron redistribution due to the redox behaviour of Cl species during charging/discharging (Supplementary Note 4). Overall, these spectroscopic results verify the presence of oxidized Cl after charging.

Although the electrochemical and spectroscopic results showed that Cl$^0$ was produced, the following uncertainties remain: how and in what configurations was Cl$^-$ fixed by coordinating with I. Therefore, we investigated the possible reaction pathways of Cl$^-$ oxidation, in which the potential energy and formula of the products were calculated along different possible oxidation reaction pathways. Starting from ICl

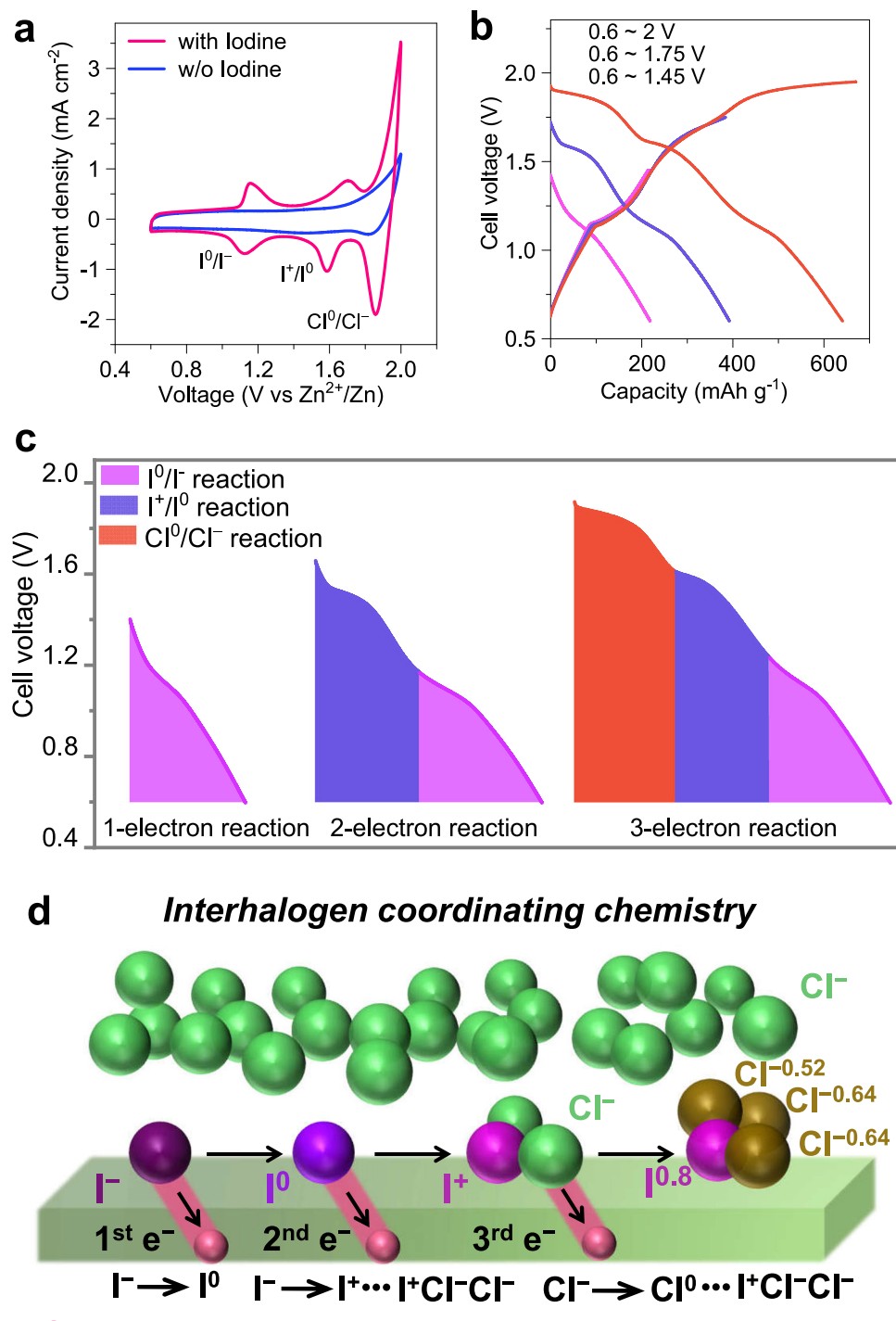

**Fig. 4 | Electrochemical investigations on the multielectrode reaction of the Cl-I positive electrode. a** CV profiles at 0.1 mV s⁻¹ to study the two functionalities of I in I-containing and I-absent electrodes. **b** GCD profiles of the Cl-I electrode operated within different potential ranges at 0.5 A $g_{I2}^{-1}$ in a Zn‖Cl-I pouch cell at 25 °C. **c** Identification of the various stages in the electrochemical reaction via cell discharge voltage analysis, in which the energy density for the 1-electron $I^0/I^-$ reaction was 233.8 Wh $kg_{I2}^{-1}$. **d** Schematics of the interhalogen coordinating chemistry to illustrate the electron flow during battery charging and electron redistributing, in which the valence states of I and Cl redox centres are marked.

and Cl⁻ as reactants, two Cl⁻ oxidation reaction compete as follows: the first produces molecular $Cl_2$ through the blue pathway, and the second is fixed by I⁺ atoms through the pink pathway, as shown in Fig. 3d. Specifically, the pink reaction pathway flows as ICl first coordinates with Cl⁻ to produce $ICl_2^-$, in which $ICl_2^-$ has also been determined to be energetically favourable[30,31]. Then, $ICl_2^-$ was further oxidized to $ICl_2$,

and it was finally stabilized by Cl⁻ in a configuration as $ICl_3^-$. A total of 3.67 eV is needed to oxidize a single Cl⁻ ion to Cl⁰ (blue curve), in which the thermodynamic energy barrier is 1.05 eV higher than that for oxidizing the $ICl_2^-$ ion to $ICl_2$ 2.62 eV. The oxidized products, i.e., Cl⁰ and $ICl_2$, are of great interest because after examining the possibility of these two products transforming into other forms through

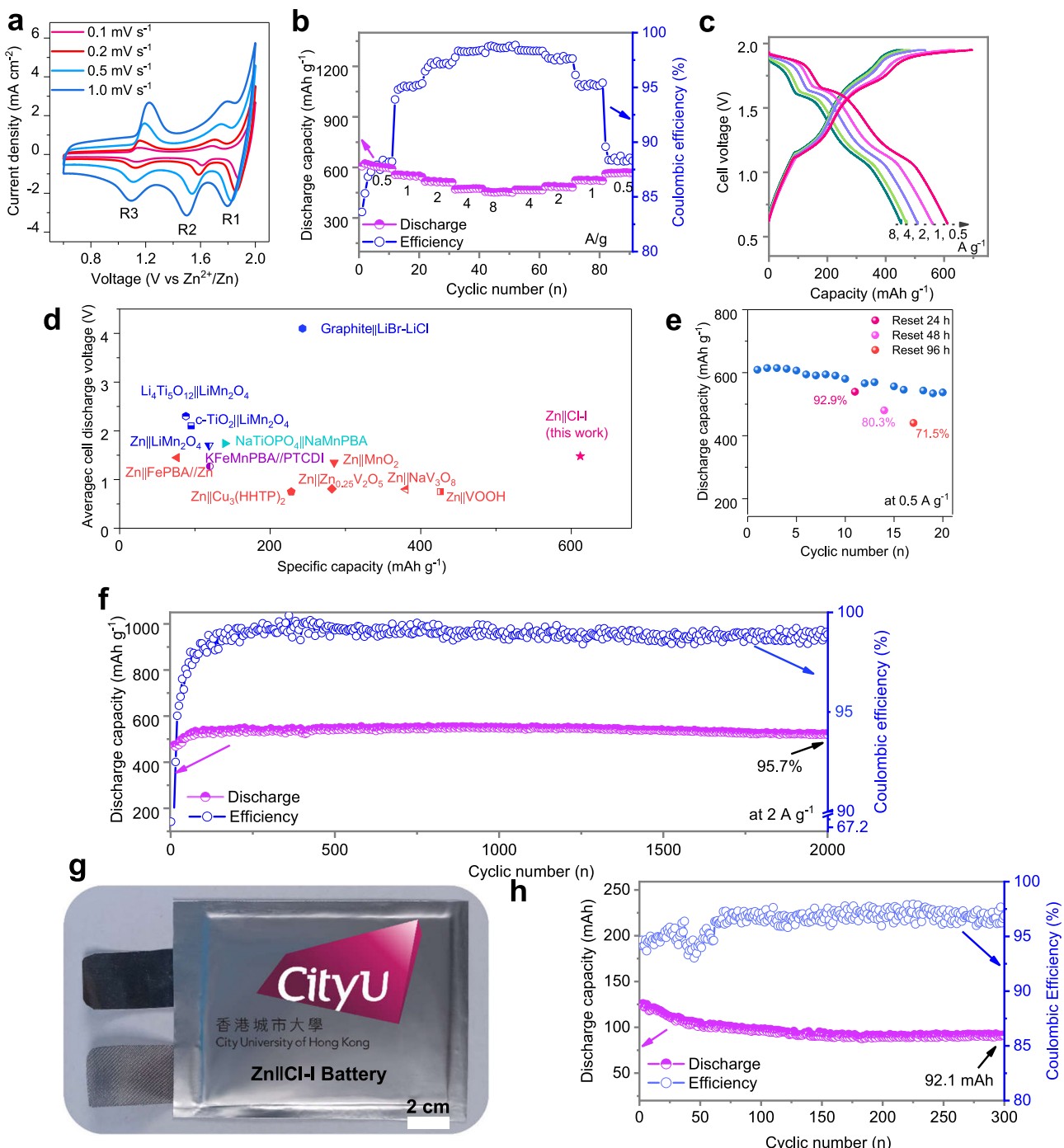

**Fig. 5 | Battery performance of various Zn∥Cl·I cells. a** CV profiles and **b** rate performance of the Zn∥Cl-I pouch cell in 30 m $ZnCl_2$ at 0.5 A $g_{I2}^{-1}$ and 25 °C. **c** Corresponding GCD curves at different currents based on the mass of iodine. **d** Comparison of the average voltage, capacity and cathode materials in different full battery systems, including $Li^+$, $Na^+$, $K^+$, and $Zn^{2+}$ ion batteries, in which the corresponding performance data are exhibited in Supplementary Table 5 for reference. The specific capacities and specific currents were based on the active materials, and it is 0.5 A $g_{I2}^{-1}$ based on the mass of iodine. **e** Storage performance evaluated by resting for different hours at a 100% state of charge at 0.5 A $g_{I2}^{-1}$. **f** The cycling stability and CE of the Zn∥Cl-I full cell at 2 A $g_{I2}^{-1}$. **g** Digital image and **h** cycling performance of the Zn∥Cl-I pouch cell at 2.5 mA $cm^{-2}$ and 25 °C.

spontaneous reactions, it was determined that they are metastable products. $Cl^0$ spontaneously produces molecular $Cl_2$ by combining with another $Cl^0$ atom, and the reaction energy needed to oxidize $Cl^-$ to $Cl_2$ is 2.5 eV. On the other hand, $ICl_2$ coordinates with $Cl^-$ to produce $ICl_3^-$, and the total energy needed to transform ICl into $ICl_3^-$ is 0.86 eV. Therefore, $ICl_3^-$ is more energetically stable than $Cl_2$ molecules, explaining why no $Cl_2$ gas is produced during charging (as shown by the pink pathway). In addition, we examined the possibilities of further oxidizing $ICl_3^-$ by losing more electrons and forming other configurations by coordinating with $Cl^-$ ions. The thermodynamic energy barriers are 4.83 eV and 3.04 eV, respectively, largely exceeding 0.86 eV as stable final products. Specifically, the pink reaction pathway flows as ICl first coordinates with $Cl^-$ to produce $ICl_2^-$, and $ICl_2^-$ has also been determined to be energetically favourable. Therefore, $ICl_3^-$ is

assigned as the final product, the cathode reaction formula is $I^+Cl^- + 2Cl^- \rightleftharpoons ICl_3^- + e^-$, and the full cell reaction formula is $2I^- + 6Cl^- + 4Zn^{2+} \rightleftharpoons Zn(ICl_3)_2 + 3Zn$. The reaction equations of each transferred electron and the corresponding reaction products are elaborated in Supplementary Table 1. According to the reaction pathway, the valence states of each atom in the $ICl_3^-$ configuration were assigned as $I^+Cl^0Cl^-Cl^-$, and the reaction equations of the full cell are presented in Supplementary Note 5. Of note, the Cl-I electrode reactions based on two redox centres are different from those based on a single redox halogen center[15,19], e.g., I and Cl, in terms of the redox species, electron transfer number, final products, and electrochemical performance (Supplementary Table 2).

The valence states of chlorine atoms in the $I^+Cl^0Cl^-Cl^-$ configuration might be unstable due to the large polarity in configurational electronegativity, and thus, the electron might redistribute of each atom in the $ICl_3^-$ configuration. To confirm this hypothesis, density functional theory analyses were conducted, and the most stabilized configuration was arrayed as a tetrahedron. The valence states of chlorine are decimals of −0.64, −0.64, and −0.52, rather than integers of 0 and -1, while the valence state of iodine is +0.8 (Fig. 3e). Thus, the final stable valence states of individual I and Cl in the $ICl_3^-$ configuration are $I^{0.8}Cl^{-0.64}Cl^{-0.64}Cl^{-0.52}$. In addition, regarding the spatial structure, the distance between the chlorine and iodine atoms is 2.56 Å for $Cl^{-0.64}\cdots I^{0.8}$ and 2.91 Å for $Cl^{-0.52}\cdots I^{0.8}$. Notably, the $ICl_3^-$ configuration is the new halogen-based cathode material, in which $Cl^0$ is stabilized by interhalogen bonding chemistry. The interacting force used to fix $Cl^0$ is chemical bonding-based interhalogen bonding rather than physical adsorption, which may accommodate enhanced interacting forces.

## Electrochemical investigations of interhalogen coordinating chemistry

After determining that ClRR can be fixed by applying I as a coordinating agent, it was determined that I can also be further exploited as a redox centre to contribute capacity when the working voltage range is widened to include the reaction potential range of I species. Generally, I can contribute to the one-electron-based capacity based on the $I^0/I^-$ conversion reaction, delivering a capacity of approximately 200 mAh $g_{I2}^{-1}$ with an average voltage of approximately 1.2 V at 25 °C[32,33]. CV profiles were measured on I-containing and I-absent electrodes, with working voltages ranging from 0.6 to 2.0 V (Fig. 4a). There were three different pairs of noticeable redox peaks for the I-containing electrode, while only one weak cathodic peak appeared at 1.83 V for the I-absent electrode. Specifically, these three cathodic peaks from higher to lower potentials at 1.84 V, 1.59 V and 1.13 V can be assigned to different reduction reactions due to $Cl^0/Cl^-$, $I^+/I^0$, and $I^0/I^-$, respectively (Fig. 4a). The CV profiles suggest that I accommodated two functionalities as the coordinating agent to fix the oxidized $Cl^0$ and the redox centre to deliver energy.

Subsequently, the electrochemical performance of the I-Cl electrode was investigated based on the GCD profiles of the three-electron reaction exhibiting three different discharging plateaus (Fig. 4b). When regulating the working potential ranges to include more different redox reactions, the GCD curves of narrow potential ranges almost overlapped with the curves of wider ones, indicating that each plateau was independently reversible. Therefore, we elaborated three reactions separately. First, for the one-electron transfer process as the $I^0/I^-$ redox electrode working within 0.6–1.45 V, the GCD profile matches well with the generally applied iodine electrode[32,33], delivering a capacity of 205 mAh g⁻¹ at 0.5 A $g_{I2}^{-1}$ and 25 °C, and the corresponding energy was 233.8 Wh kg $_{I2}^{-1}$. Second, regarding the two-electron transfer process as $I^+/I^0/I^-$ redox reactions within the full working range of 0.6–1.75 V, the above-obtained $I^0$ can be further oxidized to $I^+Cl^-$ with $Cl^-$ ions as the stabilizer. The total capacity doubled, and a different discharging voltage plateau appeared at approximately 1.59 V, in which the corresponding specific energy was 528.5 Wh kg$_{I2}^{-1}$

for the $I^+/I^0/I^-$-based redox electrode. Therefore, it achieved a 226% enhancement in specific energy, i.e., 528.5 Wh kg $_{I2}^{-1}$/233.8 Wh kg$_{I2}^{-1}$, compared to that of the $I^0/I^-$ redox electrode. Third, when the working range was widened to 0.6–2 V, the $I^+/I^0/I^-$ and ClRR can all be activated, delivering a tripled capacity compared to that of the generally applied $I^0/I^-$ one-electron electrode reaction. Of note, it was found that the capacities delivered by these three plateaus were almost identical, i.e., approximately 200 mAh $g_{I2}^{-1}$, correlating to the identical amounts of electrons transferred in each reaction at 0.5 A $g_{I2}^{-1}$ and 25 °C. Thus, the stoichiometric ratio is 1 for one I atom to fix one Cl atom and proceed with the one-electron redox reaction. This is consistent with the above calculational results, i.e., one electron is lost from the Cl species to stoichiometrically couple with one I atom. In addition, the cycling performance within different charge ranges was stable throughout varying working potential ranges (Supplementary Fig. 18).

Overall, the I-containing positive electrode shows that three-electron reactions can deliver a capacity of 612.5 mAh $g_{I2}^{-1}$, exhibiting a 289% capacity improvement over that of the one-electron-based $I^0/I^-$ redox reaction. The improved capacity and voltage jointly contribute to boosting the specific energy so that the corresponding specific energy of the Cl-I-based cathode can reach 905 Wh kg $_{I2}^{-1}$, which is 387% higher than that of the $I^0/I^-$ redox electrode generally applied in the standard Zn‖$I_2$ battery (Fig. 4c). The Zn metal cell with the I-containing positive electrode capable of a three-electron transfer reaction demonstrates improved battery performance compared to that of similar aqueous Zn-based electrochemical energy storage systems with $MnO_2$, MXene, and $V_2O_5$ cathodes (see Supplementary Fig. 19 and Table 3).

Consequently, schematics of the charging process of the Cl-I electrode are illustrated in Fig. 4d, demonstrating the electron flow and the corresponding products throughout the charge transfer process. The interhalogen coordinating chemistry between heterogeneous halogen species determines the whole process, which can be embodied by the products as ICl and $ICl_3^-$.

## Electrochemical energy storage performance of the Zn‖Cl-I battery system

Based on the previous characterizations, we assembled and tested several Zn‖Cl-I lab-scale cells capable of exploiting the $I^+/I^0/I^-$ and ClRR mechanisms. The CV measurements show the evolution of different redox peaks with increasing scanning rates from 0.1 to 1 mV s⁻¹ (Fig. 5a). The kinetics of each halogen-based reaction were analysed. The relationships between the current densities of the three cathodic peaks (*I*) and scanning rates (v) can be elaborated as $I = av^b$ (Supplementary Fig. 20), where *a* is a coefficient, a *b* value of 0.5 indicates semi-infinite diffusion behaviour, and a *b value* of 1 implies capacitive behaviour[34]. Here, all *b*-values approaching 0.5 could be assigned to halogen-based phase conversion reactions. The evolutions of three reduction peaks were carefully checked at different scanning rates, and the current density of peak R1 corresponding to ClRR increased at a slower rate compared to those of the R2 ($I^+/I^0$) and R3 ($I^0/I^-$) peaks (Fig. 5a). Therefore, the ratio of current intensities is 2 of $I_{R1}/I_{R2}$ at a low scanning rate (0.1 mV s⁻¹) and became 0.85 at 1 mV s⁻¹, which indicated the sluggish kinetic response of the R1($Cl^-/Cl^0$) reaction, as shown in close observations at 0.1 mV s⁻¹ and 1 mV s⁻¹ (Supplementary Fig. 21). In addition, the charge transfer resistances at different charged states were investigated via electrochemical impedance spectroscopy (EIS) measurements. The EIS analysis highlights the increase in the ionic transfer resistance during charging (Supplementary Fig. 22 and Supplementary Table 4), which might be assigned to the accumulation of chloride ions to retard the multiple steps of ion/electron transfer during charging of the Cl-I electrode.

The Zn‖Cl-I cell delivers a specific capacity of 612 mAh $g_{I2}^{-1}$ at 0.5 A $g_{I2}^{-1}$ and a delivered capacity of 455 mAh $g_{I2}^{-1}$ at 8 A $g_{I2}^{-1}$ and 25 °C (Fig. 5b). The corresponding GCD curves possessed three pairs of

charging–discharging plateaus as exhibited in Fig. 5c. The average cell discharge voltage and specific capacity at 0.5 A $g_{I2}^{-1}$ are well positioned compared to that of similar aqueous electrochemical energy storage systems reported in the literature (see Fig. 5d and Supplementary Table 5). Notably, the specific capacities of the Zn||Cl-I cells are calculated based on the iodine mass, while the corresponding capacity based on the total mass of iodine and AC is 226 mAh $g_{(I2+AC)}^{-1}$ at 0.5 A $g_{I2}^{-1}$, and the corresponding capacities at other specific currents are reported in Supplementary Table 6. In addition, the specific energy is 50.7 Wh $kg_{I2}^{-1}$ based on the total mass of the positive (i.e., the mass of iodine and AC without considering the Ti current collector) and negative (i.e., Zn metal) electrodes. Moreover, Zn||Cl-I cells with higher specific energy could be obtained by optimizing the iodine loading in the positive electrode (see Supplementary Note 6).

Self-discharge is an important parameter to embody the stability of the reaction products. The Zn||Cl-I cell exhibited 92.9% capacity retention after resting for 24 h and 71.5% retention after 96 h (Fig. 5e), and the corresponding GCD curves after resting are exhibited in Supplementary Fig. 23. This confirms that the stable fixation of $Cl^0$ by I species and the slight capacity losses can be assigned to the dissolution of the charged electrode materials into the electrolyte. In addition, a discharge capacity retention of 95.7% was calculated after 2000 cycles at 2 A $g_{I2}^{-1}$ at 25 °C (Fig. 5f). The Zn anode cycled in 30 m $ZnCl_2$ electrolyte showed excellent cycling stability for 200 h in symmetric cells at a current density of 1 mA $cm^{-2}$ and an areal capacity of 1 mAh $cm^{-2}$ (Supplementary Fig. 24). To further demonstrate the potential of the Zn||Cl-I battery towards practical applications, pouch cells were assembled with a cathode area of 80 $cm^2$ (10 cm × 8 cm) to demonstrate the scalability of the Cl-I electrode, as shown in Fig. 5g (Supplementary Note 7). The pouch cell delivers an initial capacity of 124.7 mAh (1.56 mAh $cm^{-2}$) at 200 mA (2.5 mA $cm^{-2}$) and a 73.8% capacity retention, i.e., 92.1 mAh, after 300 cycles (Fig. 5h).

In summary, by applying interhalogen coordinating chemistry, we demonstrated the possibility of exploiting iodine as a coordinating agent to fix oxidized $Cl^0$. A 30 m $ZnCl_2$ electrolyte was selected to suppress the OER and simultaneously inhibit I-dissolving issues, guaranteeing good cycling stability. In situ and ex situ spectroscopic and calculational results jointly verified the interhalogen coordinating chemistry between I and Cl. The final product was determined to exhibit a new configuration as $ICl_3^-$ through interhalogen bonding.

After fixing ClRR, the redox nature of I was exploited as well, obtaining two redox centres as Cl and I species with three-electron transfer. As a result, the Cl-I positive electrode, tested in combination with a Zn metal negative electrode, enables a specific capacity of 612 mAh $g_{I2}^{-1}$ at 0.5 A $g_{I2}^{-1}$, a specific energy of 905 Wh $kg_{I2}^{-1}$, and a 95.7% capacity retention after 2000 cycles at 2 A $g_{I2}^{-1}$ due to the stable interhalogen fixation.

## Methods

### Preparation of the I-containing positive electrode

First, activated carbon (AC, large surface area of ~1000 $m^2$/g, purchased from Sigma–Aldrich with the product number as 902470) was coated on titanium mesh (Ti, 50 μm thickness and 100 mesh $cm^{-2}$, purchased from Kaian metal wire mesh Co., Ltd.) as the hosting materials for the halogens I and Cl, in which the Ti mesh was selected as the current collector because of its corrosion-resistant property against chloride ions. AC was mixed with acetylene black and poly (vinylidene fluoride) (both purchased from Aladdin) at a weight ratio of 8:1:1 and then coated onto Ti mesh with a loading mass of AC of approximately 4 mg $cm^{-2}$. After drying in a vacuum oven at 80 °C, the AC host electrode was obtained. $I_2$ was synthesized by electrodeposition in a three-electrode glass cell at 25 °C with an AC electrode as the working electrode, Ag/AgCl electrode as the reference electrode, and Zn foil as the counter electrode, which were immersed in a 1 m $ZnI_2$ flooded electrolyte. A constant current density protocol was applied,

and the anodic current density was 1 mA $cm^{-2}$ and deposited for 0.5 h at 25 °C. Thus, the predeposited capacity of $I_2$ is 0.5 mAh $cm^{-2}$, which can be calculated as 2.36 mg $cm^{-2}$ based on the theoretical capacity of $I_2$ of 212 mAh $g^{-1}$. After washing with DI water and drying in a vacuum oven at 25 °C, the I-containing electrode was obtained.

### Electrochemical characterizations

A 30 m $ZnCl_2$ electrolyte was prepared through dissolving 30 mol $ZnCl_2$ salt in 1 kg deionized water at 25 °C. The 30 m $ZnCl_2$ configuration (100 for μL for 1 $cm^2$ size cell) was applied for all cell configurations to investigate interhalogen redox chemistry and performance, while other concentrations, e.g., 1, 2, 5, 10, 20 m $ZnCl_2$, were applied for comparative study as described in the main text. The Zn||Cl-I pouch cells were configured by applying an I-containing electrode sized 1 $cm^2$ (1 cm × 1 cm) as the cathode, where Zn foil (1 cm × 1 cm, 99.99% purity, purchased from Aladdin) with a thickness of 50 μm was applied as the anode. Whatman GF/F glass microfiber discs saturated with specific electrolytes are utilized as separators between the cathode and anode. Large-sized (10 cm × 8 cm) single-side coated I-containing positive electrodes and Zn negative electrodes were utilized to assemble pouch cells in an air environment. An ethylene-ethyl acrylate sheet was applied as the encapsulation layer for the pouch cells. Five cells were tested for a single electrochemical experiment for the CV, rate and cycling performance. The blank AC electrode was applied for comparison with the AC electrode containing deposited $I_2$, while the specific gravimetric capacities of I-containing and I-absent electrodes were calculated based on the $I_2$ mass to directly study the capacity contributed by $I_2$. The electrochemical tests of the batteries are carried out by a CHI 760E electrochemical working station and Land 2001A battery testing system, in which cyclic voltammetry (CV) and galvanostatic charge/discharge are conducted. Linear sweep voltammetry (LSV) was performed with Ti foil as the working electrode and Zn as the anode, and the anodic scan was initiated at the voltage of the open circuit potential at a scanning rate of 0.5 mV $s^{-1}$. Electrochemical impedance spectroscopy to obtain the Nyquist plots was carried out with an AC perturbation signal of ±5 mV and a frequency range from 0.1 Hz to 100 kHz to measure the selected quasistationary potentials during the anodic CV scan. All these electrochemical tests were carried out at an environmental temperature of 25 °C, and no climatic/environmental chamber was used. Since the mass of $Cl^-$ ions involved in the electrode reactions is always changing, it cannot represent the real-time capacity to count the varying mass of $Cl^-$ anions; thus, the specific capacity and specific energy of the cells are calculated based on the mass of iodine.

### Material characterizations

The Raman spectra were collected for different electrolytes using a PerkinElmer Raman 400 F Spectrometer equipped with a 532 nm NIR laser, and the measurement was conducted by focusing the laser light onto the electrolyte samples in a quartz tube. The ex situ Raman analysis of electrode evolution was carried out in an electrolyte flooded cell in two-electrode Zn||Cl-I configurations sealed by quartz glass, and the data were collected at different states of charge and discharge at 25 °C. X-ray photoelectron spectroscopy (XPS) measurements were conducted by a PHI Model 5802, and UV–vis spectra were collected by a Shimadzu UV 3600 UV/visible/IR spectrophotometer. The electrode samples at different states were washed with DI water and then maintained under vacuum overnight to conduct XPS and UV–vis measurements. The gas concentration detection of gaseous $Cl_2$ was measured by a Kallu Electronics detector based on a home-designed cell, as presented in Supplementary Fig. 3. The detailed measuring protocols are described as follows: first, before charging the cell, $N_2$ was fully injected to guarantee that the cell head space was filled with $N_2$. Second, during charging, the valves connected to the two tubes were closed, and the cell was charged at 0.5 A $g^{-1}$ to specific voltages

and maintained at the voltage point for 90 s to sufficiently obtain the oxidized products. Finally, the valve of Tube 2 was opened to extract the gas in the cell head space for $Cl_2$ detection, and the valve of Tube 1 remained closed. Three steps are used to detect the $Cl_2$ concentration at one specific voltage. All voltage points (correlated to the voltages in Fig. 1c in the main text) were performed accordingly for gaseous $Cl_2$ detection.

## Computational method

All MD simulations were implemented in the open-source code Large-scale Atomic/Molecular Massively Parallel Simulator (LAMMPS) [1-SI] with the nonbonded interatomic potentials described by the Lenard-Jones (LJ) force field. The $TIP4P_{EW}$ model was chosen to simulate water molecules in the system. The parameters for Zn ions and Cl ions were obtained from Merz's works of fitting LJ parameters for aqueous +2 metal cations [2-SI] and monovalent ions [3-SI], respectively. Systems of boxes with $C_{ZnCl_2}$ of 1 m, 10 m, 20 m, and 30 m were constructed to model the aqueous electrolyte with different concentrations. The compositions of these systems are shown in Supplementary Table 7. First, these systems were heated with NVT ensembles for 5 ns with different temperatures (from 300 K to 800 K) varying with the concentrations and then relaxed with an NPT ensemble for 45 ns to ensure equilibrium. Finally, an NVE run for 2 ns was performed to collect and analyse the data. The H-bond number was calculated using the Hydrogen Bond Anaysis code [4-SI] in the MD Analysis package [5-SI] with the criteria that the distance of donor O and accepter O was <3.5 Å, and the angle of $_{(donor)}O\text{-}H{\cdots}O_{(acceptor)}$ was >150°. Quantum chemistry calculations were carried out through the Gaussian 09 software package [6-SI]. The B3LYP/SDD basis set was used for Zn, and the B3LYP/6-311 + G(d) basis set was used for H, O, and Cl [7,8-SI]. The visualization of the simulation process was realized by VMD software [9-SI].

The electronic structure calculations, including the geometries, energies, and frequencies of all the stationary points (the reactants, transition states (TSs), products) were performed by the GAUSSIAN 09 program [10-SI]. Density functional theory calculations (DFT) were carried out by using the B3LYP (Becke's three-parameter nonlocal-exchange functional [11-SI] with the gradient correction of Lee, Yang, and Parr [12-SI]) method together with the aug-cc-pVTZ basis set for the Cl atom and the effective core potential (ECP) for the I element. The references listed in the Computational Method are numbered from 1 to 12 in Supplementary Reference List at the end of Supplementary Information.

## Data availability

All the data generated during this research are included in this article and its Supplementary information. All relevant data are available from the authors, and requests for datasets should be addressed to Y.W. or J.F., C.Z.

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

## Acknowledgements

This research was supported by the National Key R&D Program of China under Project 2019YFA0705104. The work was also partially sponsored by GRFs under Project CityU 11304921. The work described in this paper was substantially supported by a fellowship award from the Research Grants Council of the Hong Kong Special Administrative Region, China (Project No. CityU PDFS2122-1S05).

## Author contributions

G.L. and C.Z. designed the study. C.Z., Y.W., and J.F. supervised the experiments. G.L., A.C., J.Z., Q.L., Z.H., X.L., X.W., B.X., X.J., S.B. conducted the electrochemical and spectroscopic characterizations and analyzed the data. B.L., Y.W., and J.F. conducted the theoretical calculations. All authors discussed the results and commented on the manuscript.

## Competing interests

The authors declare no competing interests.
