## [Peer Review File · Nature Communications]

REVIEWER COMMENTS

Reviewer #1 (Remarks to the Author):

The authors have addressed my concerns, and have made changes accordingly. It can be accepted as is

Reviewer #3 (Remarks to the Author):

This manuscript reported a Cl-I ion batteries. Authors used I element to fix oxidized ClO via forming ICl₃⁻. The highly concentrated 30 m ZnCl₂ electrolyte was used to achieve better electrochemical performances for Cl/I - Zn ion batteries. Rechargeable chlorine battery has been well investigated (Nature 596 (2021) 525). Meanwhile, rechargeable iodine batteries have also been extensively studied. Therefore, using I and Cl as cathodes is not innovative. Consequently, the novelty of this manuscript is not high enough for publication in Nature Communications. The issues of this manuscript are outlined below.

1. Both Cl and I contribute capacity to the Cl-I hybrid batteries. The authors should calculate capacities based on the weight of both I and Cl. Because Cl contributes 1 electron per reaction, which is only about 1/3 of the total capacity. The actual capacity of the Cl-I electrode could be significantly lower than that claimed in the manuscript.
2. The authors should provide redox reaction equations of the Cl/I-Zn ion battery.
3. In terms of electrolyte selection, authors tested different concentrations of ZnCl₂ electrolytes. The solvation structure of the electrolytes should be analyzed. The electrochemical stability window of each electrolyte should be tested and presented.
4. The dynamics of the concentrated electrolytes should be investigated using molecular dynamic simulation.
5. When construct battery cells, the authors used activated carbon as the host electrode materials. ClO could be absorbed in activated carbon, which could also contribute to the capacity of the battery.
6. According the 3-electrons reactions in Cl-I batteries described in the manuscript, Chlorine contributes 1 electron reaction and Iodine contributes 2 electrons reactions. When calculate the specific capacity, the total mass of Cl and I should be used for calculation.

7. Authors should clearly describe that the specific capacity presented in Fig. 5b, c, e and f is calculated based on the mass of I or the total mass of Cl+I? If they are based on the mass of I, it is not correct.
8. Why did authors define 0.5Ag-1 as 1 C rate?
9. The detailed and in-depth electrochemical reactions of Cl-I electrode should be provided for clearly understanding the electrochemical redox mechanism of the Cl-I ion batteries.
10. Why was Ti mesh used as the current collector? Ti mesh is not suitable for practical application because it is quite expensive.
11. In the Cl-I ion batteries, the cathode consists of activated carbon and I. If count the mass of activated carbon, what is the specific capacity of Cl-I anodes?
12. The Cl-I electrodes were prepared by electrochemically deposit I₂ on activated carbon matrix. Is this method feasible for preparing electrodes with large area.
13. Overall, it is suggested to submit this manuscript to specialized journals such as Chem, Angew. Chem. Or JMCA.

Dear Reviewers:

Thanks a lot for your constructive comments and suggestions concerning our manuscript entitled *“Fixing reversible Cl/Cl⁰ conversion reaction and enabling high-energy interhalogen-based Cl-I hybrid cathode with three-electrons transfer”*. Those comments are all valuable and very helpful for improving the quality of our paper. We have carefully studied your comments and substantially revised our manuscript which we hope to meet with approval.

We provide a point-by-point response below and the involved changes had been highlighted in the revised manuscript, where the original comments are shown in black and our responses are shown in blue.

=====

Reviewer #1: The authors have addressed my concerns, and have made changes accordingly. It can be accepted as is.

Answer: We are delighted on having addressed your concerns and thank you for the positive comments on our manuscript.

Reviewer #3:

This manuscript reported a Cl-I ion batteries. Authors used I element to fixe oxidized Cl⁰ via forming ICl₃-. The highly concentrated 30 m ZnCl₂ electrolyte was used to achieve better electrochemical performances for Cl/I - Zn ion batteries. Rechargeable chlorine battery has been well investigated (Nature 596 (2021) 525). Meanwhile, rechargeable iodine batteries have also been extensively studies. Therefore, using I and Cl as cathodes is not innovative. Consequently, the novelty of this manuscript is not high enough for publication in Nature Communications. The issues of this manuscript are outlined below.

Answer: The rechargeable Cl battery based on Cl-redox electrode (which you listed as **Nature 2021, 596, 525**, here denoted as **Paper 1**), is reported to deliver high energy density, suggesting the Cl-redox electrodes are promising for high-energy batteries. However, exploiting the Cl-based redox electrode was realized based on the physical adsorption, while fixing Cl⁻/Cl⁰ based on the chemical bonding has not been realized. The key findings of our study are to exploit the Cl-based redox chemistry, rather the only iodine batteries. We agreed with your judgment on the “*rechargeable iodine batteries have also been extensively studies*”, but the main point is to fix the reversible Cl⁻/Cl⁰ conversion reaction by I, where the I could fix the reversible Cl⁻/Cl⁰ conversion reaction and meanwhile deliver capacity as a redox center. Compared to the general host carbon materials as Cl⁻/Cl⁰ host based on physical adsorption with capacitive capacity, the I species as fixing agents not only accommodated a much-improved capacity of the Cl⁻/Cl⁰ electrode, but also itself proceeded two-electron transfer redox reactions for higher capacity. This bifunctionality of I was different from the conventional iodine batteries. To make it more comprehensive, we have compared our work with the listed work, *i.e.*, Paper 1, from 4 aspects to present the novelty of our interhalogen-based high-energy electrodes. We acknowledge that some parts of our experiment *per se* are not newly reported in this manuscript, but this fact alone should not preclude our paper from publishing since every new research must base on previous works. Therefore, we believe our work is innovative to develop interhalogen-based high-energy electrodes. We have added more careful discussions about the difference between the listed **Paper 1** and our work, please see the red-fond section on **Page 14** and **Table S2** in Supplementary Information.

1. Both Cl and I contribute capacity to the Cl-I hybrid batteries. The authors should calculate capacities based on the weight of both I and Cl. Because Cl contributes 1 electron per reaction, which is only about 1/3 of the total capacity. The actual capacity of the Cl-I electrode could be significantly lower than that claimed in the manuscript.

Answer: Thank you for your valuable and helpful comments. The mass of Cl⁻ was not counted in the total mass when calculating the specific capacity, because the Cl⁻ ions are resourced from the electrolyte, moving into the cathode side during the charging process and reversing back into the electrolyte during discharging. The mass of Cl⁻ ions involving the electrode reactions are always changing, and thus, it could not represent the real-time capacity based on the varying mass of Cl⁻ anions.

It should be noted that when the capacity calculation involves the anions from the electrolyte, it is a general protocol not to count the ion masses resourcing from the electrolyte. For a representative example, the cathode capacity to host the AlCl₄⁻ anions in the aluminum ion battery was calculated based on the mass of the host active materials, such as graphite (*Nature* **520**, 324, 2015, *Nat. Commun.* **8**, 14283, 2017), and the TiS_x (*Sci. Adv.*, **7**, eabg6314, 2021), without counting the AlCl₄⁻ ions resourcing from the electrolyte. In addition, regarding the other anions-based batteries, the specific capacity was also calculated based on the masses of the graphite cathode without counting the masses of the anions from the electrolyte (*Nat. Commun.* **12**, 3106, 2021).

We have clarified the specific capacity of the electrode is calculated based on the mass of fixing agents as iodine (red-fond section on Page 5). Since we have experimentally identified that the Cl contributes 1 electron per reaction and it was found that the capacity delivered by the three plateaus were almost identical (Fig. 4b), the total output capacity and specific capacity delivered from the Cl redox reaction can be reflected by using the mass of I. Thus, we believe it is reasonable to use the mass of I as active materials to calculate the specific capacity of the Cl-I electrode.

We are sorry for having not discussed this point in our original manuscript, and according to your valuable comments, we have added the discussions about the reason to only take the I as the mass to calculate the specific capacity, please see the red-fond section on Page 5 and Supplementary Note S1 in Supplementary Information.

2. The authors should provide redox reaction equations of the Cl/I-Zn ion battery

Answer: Thank you for your insightful comments. At the cathode side, the chemical reaction equation can be provided by a detailed electrochemical mechanism for fixing Cl⁰ by I₂. It is a three-electrons reaction with three individual reaction equations, where the I⁻ and Cl⁻ can be oxidized within different charging voltage ranges. In specific, the I can be oxidized in sequence from I⁻ to I⁰ and to I⁺ in the equation as $I^- \rightleftharpoons I^0 + e^-$, and $I^0 + Cl^- \rightleftharpoons I^+Cl^- + e^-$. I⁺Cl⁻ can be stabilized by Cl⁻ ion and form the more stabilized products as

ICl_2^- in the equation as $\text{I}^+\text{Cl}^- + \text{Cl}^- \rightleftharpoons \text{ICl}_2^-$. Then, the ICl_2^- can be further oxidized and stabilized by Cl^- to form a stable configuration as ICl_3^- in the equation as $\text{ICl}_2^- + \text{Cl}^- \rightleftharpoons \text{ICl}_3^- + \text{e}^-$. Thus, the cathode reaction can be expressed as $\text{I}^- + 3\text{Cl}^- \rightleftharpoons \text{ICl}_3^- + 3\text{e}^-$. As for the Zn anode, the reaction equation can be expressed as $\text{Zn}^{2+} + 2\text{e}^- \rightleftharpoons \text{Zn}$. Thus, the full reaction equations of the Cl/I-Zn ion battery is

We are sorry for having not discussed this point in-depth in our original manuscript. According to your suggestion, we have explained the above-mentioned reaction equations at the cathode and the full cell, please check to see the red-fond section on Page 14 and Table S1 and Note S5 in Supplementary Information.

3. In terms of electrolyte selection, authors tested different concentrations of ZnCl_2 electrolytes. The solvation structure of the electrolytes should be analyzed. The electrochemical stability window of each electrolyte should be tested and presented.

Answer: Thank you for your valuable and helpful comments. According to your suggestion, the solvation structures of the electrolytes, *i.e.*, 1 M, 10 M, 20 M and 30 M ZnCl_2 electrolytes, were analyzed by the molecular dynamic (MD) simulation. In specific, the snapshots of simulation boxes of various concentrations as well as the zoom-in images were displayed about the corresponding major solvation structure of Zn^{2+} and their proportions in the system, where the major solvation structure of Zn^{2+} and their proportion were demonstrated. In addition, radial distribution functions (RDFs, $g(r)$) and coordination number ($n(r)$) of Zn-Cl and Zn-O were plotted to further understand the coordination structure of Zn^{2+} ions. Along C_{ZnCl_2} increases, the coordination number of H_2O decreases from about 5.6 to 1.6, while that of Cl^- increases from about 0.4 to 3.8. Therefore, the Cl^- will compete with H_2O to coordinate Zn^{2+} and gradually replace the highly active coordinated H_2O at high concentration, which suppresses the reactivity of water and help alleviate the water-induced parasitic reactions. The ratio distributions of corresponding solvation structures in various systems are calculated by setting the radius of the first solvation shell (*i.e.*, the distance of the first valley in the RDF plot) as the cutoff distance.

At low concentrations, as the 1 M ZnCl_2 electrolyte, $\text{Zn}(\text{H}_2\text{O})_6^{2+}$ is the major solvation structure, which takes up 64.44% Zn^{2+} in the system, indicating the high reactivity of water towards Zn^{2+} . As the concentration increases, the Cl^- ions would gradually dominate the first solvation shell with the $\text{Zn}(\text{H}_2\text{O})_3\text{Cl}_3^-$ and $\text{Zn}(\text{H}_2\text{O})_4\text{Cl}_2$ becomes dominant at 10 M and 20 M. However, when the concentration increased to 30 M, the proportion of the $\text{Zn}(\text{H}_2\text{O})_3\text{Cl}_3^-$ and $\text{Zn}(\text{H}_2\text{O})_4\text{Cl}_2$ obviously declines, while $\text{Zn}(\text{H}_2\text{O})_2\text{Cl}_4^{2-}$ grows to occupy 32.07% Zn^{2+} in the system. It should be noted that the four coordinated

ZnCl_4^{2-} occur when concentration increases to 20 M and become even more at 30 M, which implies the stronger binding between Zn^{2+} and Cl^- .

Following your comments, we have also tested the electrochemical stability window of each electrolyte from 1 M to 30 M. In specific, the OER set-off potential is located at around 1.95 V in the low-concentration electrolyte (1 M ZnCl_2), while OER was suppressed to 2.1 V in the high concentration electrolyte (30 M ZnCl_2). The Cl evolution potential was decreased from 2.13 V of the 1 M ZnCl_2 electrolyte to 1.98 V of the 30 M ZnCl_2 electrolyte. Thus, the water activity of the OER was suppressed in the higher ZnCl_2 electrolyte to include the Cl-evolution reactions.

We are sorry for having not discussed the solvation structure in-depth in our original manuscript and the solvation structure of the four tested ZnCl_2 was analyzed according to your suggestion. The above-mentioned solvation structure, the RDF analysis, and the ratio distribution of different solvation structures were exhibited in Supplementary Fig. S9, Fig. S10, Fig. S11, and Fig. S12, respectively. In addition, the electrochemical windows of different electrolytes were exhibited in Fig. S5. Please check to see the red-fond section on Page 7, Page 8, and Page 9 in the main text and the Calculation Section in Supplementary Information.

4. The dynamics of the concentrated electrolytes should be investigated using molecular dynamic simulation.

Answer: Thank you for your valuable and helpful comments. Following your comments, to analyze the dynamics of the electrolytes, *i.e.*, 1 M, 10 M, 20 M and 30 M ZnCl_2 electrolytes, the binding energy of one water molecule with the rest structure of $\text{Zn}(\text{H}_2\text{O})_6^{2+}$, $\text{Zn}(\text{H}_2\text{O})_4\text{Cl}_2$, $\text{Zn}(\text{H}_2\text{O})_2\text{Cl}_4^{2-}$ was estimated by quantum chemistry calculations to explore the evolution of dominant solvation structures in low concentration electrolyte as $\text{Zn}(\text{H}_2\text{O})_6^{2+}$ and the high concentration electrolyte systems as $\text{Zn}(\text{H}_2\text{O})_4\text{Cl}_2$, $\text{Zn}(\text{H}_2\text{O})_2\text{Cl}_4^{2-}$. In specific, the binding strength of one water molecule of $\text{Zn}(\text{H}_2\text{O})_6^{2+}$ as $-34.77 \text{ kcal mol}^{-1}$ is much stronger than that for $\text{Zn}(\text{H}_2\text{O})_4\text{Cl}_2$ and $\text{Zn}(\text{H}_2\text{O})_2\text{Cl}_4^{2-}$ as $-3.09 \text{ kcal mol}^{-1}$ and $-2.77 \text{ kcal mol}^{-1}$, respectively, indicating that water molecules stay much more stable in $\text{Zn}(\text{H}_2\text{O})_6^{2+}$. In addition, the very low water binding energy of $\text{Zn}(\text{H}_2\text{O})_4\text{Cl}_2$ and $\text{Zn}(\text{H}_2\text{O})_2\text{Cl}_4^{2-}$ and the small difference of $0.32 \text{ kcal mol}^{-1}$ between them imply that water molecules in these structures are easy to be pulled out and replaced by Cl^- in high concentrated electrolytes. Therefore, along the C_{ZnCl_2} increases, the proportion of the $\text{Zn}(\text{H}_2\text{O})_4\text{Cl}_2$ decreases while the $\text{Zn}(\text{H}_2\text{O})_2\text{Cl}_4^{2-}$ increases to finally become dominant in 30 M ZnCl_2 electrolyte.

On the other side, the reactivity of water can be also suppressed by breaking the hydrogen bonds (H-bonds) network in the aqueous electrolyte along the C_{ZnCl_2} increases. Therefore, H-bonds in different

simulation boxes are shown to explore the impact of C_{ZnCl_2} on the H-bonds formed between water and water, from which we can intuitively see the decline of the H-bonds number from low concentration to high concentration. To be specific, we calculated the average hydrogen bond number formed by one water molecule for each concentration. At low concentrations (1 M Zn_2Cl), the average H-bonds number of one water molecule is 1.21, while as the concentration increases, the H-bonds number decreases to 0.53, 0.17, and 0.08 in the 10 M, 20 M, and 30 M system, respectively. These results demonstrate that the H-bonds network is broken as C_{ZnCl_2} increases, which also contributes to the lower reactivity of water in high C_{ZnCl_2} case. It is consistent with the electrochemical stability window test that the OER was suppressed to enable the stable realization of Cl⁻ ion redox reaction.

We are sorry for having not discussed the electrolyte dynamics in-depth in our original manuscript. The above-mentioned binding strengths between one water molecule and Zn^{2+} ion of the different solvation structures are exhibited in Supplementary Fig. S12, where the snapshots of hydrogen bonds in simulation boxes of different systems are also exhibited in Supplementary Fig. S13. In addition, the average number of hydrogen bonds number per H_2O in systems of different concentrations is exhibited in Supplementary Fig. S14. Please check to see the red-fond section on Page 8, Page 9 in the main text and the Calculation Section, and the Supplementary Note S3 in Supplementary Information.

5. When construct battery cells, the authors used activated carbon as the host electrode materials. ClO could be absorbed in activated carbon, which could also contribute to the capacity of the battery.

Answer: Thank you for your valuable and helpful comments. We tested the process of Cl_2 evolution reaction (CIER), which is directly related to the working potential. In specific, the CIER will be triggered once the charging potential exceeds 1.95 V based on the linear sweep voltammetry (LSV) results (Fig S5). The result is consistent with the results of gaseous Cl_2 evolution detection, where the Cl_2 gas evolution of bare activated carbon (AC) electrode began to be obvious after holding 90 seconds at 1.95 V (Fig. 1c). We have further tested the capacity of bare AC and there is no obvious discharging plateau of Cl^0/Cl^- conversion reaction, where the capacity can be regarded as capacitive ion adsorption of rather than the Cl^0 involved redox reaction (Fig. S1 and Fig. 1b). Overall, there are capacitive capacity dominated and slight Cl^0 to contribute the total capacity of the activated carbon.

To be more specific, the GCD profile of bare AC electrode at $0.5 A g^{-1}$ was tested and the corresponding specific capacity_{AC} is $14.6 mAh g^{-1}$ calculated based on the I_2 mass as active materials (Figure S1). Then, we recalculated the specific capacity of the halogen electrode based on the total mass of iodine and AC. Since the mass of electrodeposited iodine is $2.36 mg cm^{-2}$ and the mass of AC is $4 mg cm^{-2}$, the specific capacity would be

$$C_T = C_I * 2.36 / (2.36 + 4) = 0.37 C_I$$

where the C_I is the capacity calculated based on the only mass of iodine and C_T is the capacity based on the total mass of iodine and AC. Where one comparative Table S5 was clarified to elaborate their difference in the calculations. In addition, after obtaining the interhalogen chemistry, we emphasized the importance of increasing the loading of the active halogen species or reducing the mass of hosting materials to realize higher capacity.

We are sorry for having not discussed this point in our original manuscript, and according to your valuable comments, we have added the specific capacity based on the total mass of iodine and AC, please see the red-fond section on Page 5, 6, 19 in main text and Fig. S1, Table S5 and Note S6 in Supplementary Information.

6. According the 3-electrons reactions in Cl-I batteries described in the manuscript, Chlorine contributes 1 electron reaction and Iodine contributes 2 electrons reactions. When calculate the specific capacity, the total mass of Cl and I should be used for calculation.

Answer: Thank you for your valuable and helpful comments. We have specified that the specific capacity is calculated based on the mass of fixing agents such as iodine. The Cl^- ions are resourced from the electrolyte, moving into the cathode side during the battery charging, and the Cl^- ions would return back into the electrolyte during discharging. The mass of Cl^- ions involving the electrode reactions are always changing, and thus, it could not represent the real-time capacity to count in the varying mass of Cl^- anions. During battery operation, the mass value of the total mass of Cl and I is constantly changing, while the mass of I is fixed. Thus, we only used the mass of I, which is regarded as a constant mass value, and it can clearly present the capacity of the Cl-I electrode. It should be noted that, when the capacity calculation involving the anions from the electrolyte, it is a general protocol not to count the ion masses resourcing from the electrolyte. For a representative example, the cathode capacity to host the $AlCl_4^-$ anions in the aluminum ion battery was calculated based on the mass of the host active materials, such as graphite (*Nature* **520**, 324, 2015, *Nat. Commun.* **8**, 14283, 2017), and the TiS_x (*Sci. Adv.*, **7**, eabg6314, 2021), without counting the $AlCl_4^-$ ions resourcing from the electrolyte. In addition, regarding the other anions-based batteries, the specific capacity was also calculated based on the masses of graphite cathode without counting the masses of the anions from the electrolyte (*Nat. Commun.* **12**, 3106, 2021). On the other side, since we have experimentally identified that the Cl contributes 1 electron per reaction and it was found that the capacity delivered by these three plateaus were almost identical (Fig. 4b), the total output capacity and specific capacity delivered from the Cl redox reaction can be reflected by using the mass of I. Thus, it is reasonable to use the mass of I to calculate the specific capacity of the Cl-I electrode.

We are sorry for having not discussed this point in our original manuscript, and according to your valuable comments, we have added the discussions about the reason to only take the I as the mass to

calculate the specific capacity, please see the red-fond section on Page 5 and Supplementary Note S1 in Supplementary Information.

7. Authors need to clarify that the specific capacity presented in Fig. 5b, c, e and f is calculated based on the mass of I or the sum of Cl+I? If they are based on the mass of I, it is not correct.

Answer: Thank you for your valuable and helpful comments. Because the Cl⁻ anions involving the electrode reactions are shuttling between the electrolyte and electrode, the corresponding real-time mass of the sum of I and Cl in the interhalogen electrode mass is changing. Therefore, the specific energy density cannot be reflected by using the varying mass of the sum of Cl and I. On the contrary, due to the mass of iodine being constant, the real-time capacity of Cl-I redox electrode can be accurately reflected. Thus, these specific capacities are calculated according to the mass of iodine, which is a general protocol to calculate the anion-involved electrode based on the active materials of the cathode masses, which has been demonstrated in the following literature as *Nature* **520**, 324, 2015; *Nat. Commun.* **8**, 14283, 2017; *Sci. Adv.*, **7**, eabg6314; *Nat. Commun.* **12**, 3106, 2021.

We are sorry for having not discussed this point in our original manuscript, and according to your valuable comments, we have added the discussions about the reason to only take the I as the mass to calculate the specific capacity, please see the red-fond section on Page 5, and Supplementary Note S1 in Supplementary Information.

8. What is the reason to define 0.5Ag-1 as 1 C rate?

Answer: Thank you for your valuable and helpful comments. We tested the specific capacity under different current densities, where the current increased multiple. It is the general protocol to define the smallest current density and the maximum discharge capacity with the discharging of the cell taking 1 hour as the 1 C-rate, as demonstrated in the representative literature as *Nature Nanotech.* **6**, 277, 2011 and *J. Am. Chem. Soc.* 2017, 139, 9775.

In our study, 0.5 A g⁻¹ is the smallest current density among our tested current density values, while corresponding to the largest specific capacity of 612 mAh g⁻¹ with the discharging of the cell taking 1 hour. It is close to the theoretical capacity of 636 mAh g⁻¹ based on the 3-electron transfer reaction of the interhalogen redox chemistry. Thus, we take the general protocol to define the tested smallest current density (0.5 A g⁻¹) as 1 C rate, and then 1 A g⁻¹ as 2 C, 1.5 A g⁻¹ as 3 C and so on, as shown in Fig. 5b and Fig. 5c.

We are sorry for having not discussed this point in our original manuscript, and according to your valuable comments, we have added the discussions about the reason to only take the defined 0.5 A g⁻¹ as 1 C rate. Please see the red-fond section on Page 19 and the Method section on Page 24.

9. The detailed and in-depth electrochemical reactions of Cl-I electrode should be provided for clearly understanding the electrochemical redox mechanism of the Cl-I ion batteries.

Answer: Thank you for your valuable and helpful comments. According to your valuable comments, chemical reaction equations are provided to elaborate the detailed electrochemical mechanism for the electrochemical reactions of Cl-I electrode. It is a three-electrons reaction with three individual reactions, where the I⁻ and Cl⁻ can be oxidized within different charging voltage ranges. In specific, the I can be oxidized in sequence from I⁻ to I⁰ and to I⁺ in the equation as $I^- \rightleftharpoons I^0 + e^-$, and $I^0 + Cl^- \rightleftharpoons I^+Cl^- + e^-$. Then, the I⁺Cl⁻ can be stabilized by Cl⁻ ion and form the more stabilized products as ICl₂⁻ in the equation as $I^+Cl^- + Cl^- \rightleftharpoons ICl_2^-$. Lastly, the ICl₂⁻ can be further oxidized and stabilized by Cl⁻ to form a stable configuration as ICl₃⁻ in the equation as $ICl_2^- + Cl^- \rightleftharpoons ICl_3^- + e^-$. It can be reflected by the energy profiles of these reaction pathways and reaction products (Fig. 3d). In addition, all reactants and products are consistent with spectroscopic results about the reaction species. Thus, the full reaction equations of the Cl/I-Zn ion battery is

We are sorry for having not discussed this point in our original manuscript, and according to your valuable comments, these reaction equations are concluded and provided in Table S1, please see the red-fond section on Page 14 and Table S1 in Supplementary Information

10. Why Ti mesh was used as the current collector? It is not practical to use Ti mesh to construct batteries in industry due to high cost of Ti.

Answer: Thank you for your valuable and helpful comments. We agree Ti is expensive, but our research focuses on the new chemistry resulting in a high-performance battery. We are not claiming that this technology can be commercialized in near future. We admit our research is at a fundamental level, similar to many other reports, where the Ti-based current collectors and the electrode electrodeposition method have been widely utilized in battery research and electrochemical research, such as the Ti current collector utilized in *Nat. Energy* **1**, 16129 2016, and *Energy Storage Mater.*, **18**, 199, 2019. We have attempted to utilize cheap stainless steel as current collectors, but the severe corrosion effect of chloride ions on stainless steel leads to unstable results. Thus, we turned to choose non-corrosive Ti metal as the current collector to support our investigations on the interhalogen chemistry of halogen electrodes. We will try to solve such a cost issue of the halogen-electrode in our following research. Here, we want to note that we did not claim in our manuscript that our Cl-I electrode could be applied to the industry at the current stage.

We are sorry for having not discussed the reason to use the Ti mesh in our original manuscript and we have added these discussions according to your valuable comments, please see the red-fond section of Methods on Page 23 in the main text.

11. In the Cl-I ion batteries, the cathode consists of activated carbon and I. If count the mass of activated carbon, what is the value of specific capacity of Cl-I anodes?

Answer: Thank you for your valuable and helpful comments. Considering our main research target is to resolve the multi-electron transfer interhalogen chemistry, we took the iodine mass as active materials in the electrode to calculate the specific capacity, rather than including the activated carbon (AC) as active electrode materials to contribute capacity in our original manuscript. According to your suggestion, we tested the capacity delivered by the bare AC, where the dominant capacitive capacity and slight Cl^0 -redox capacity together contributed to the output capacity of the bare AC host within the tested voltage range as 1.7 to 2 V (Fig. 1c and Fig.S5c). The GCD profile of bare AC electrode without I_2 at 0.5 A g^{-1} was tested and the corresponding specific capacity_{AC} is 14.6 mAh g^{-1} (Figure S1). Of note, this specific capacity_{AC} here is calculated based on the I_2 mass. According to your suggestion, we have recalculated the specific capacity of the halogen electrode based on the total mass of iodine and AC. Since the mass of electrodeposited iodine is 2.36 mg cm^{-2} and the mass of AC is 4 mg cm^{-2} , the specific capacity would be

$$C_T = C_I * 2.36 / (2.36 + 4) = 0.37 C_I$$

where the C_I is the capacity calculated based on the only mass of iodine and C_T is the capacity based on the total mass of iodine and AC. Where one comparative Table S5 was clarified to elaborate the specific capacity at different current densities based on the total mass of iodine and AC. In addition, after obtaining the interhalogen chemistry, we emphasized the importance of increasing the loading of the active halogen species or reducing the mass of hosting materials to realize higher capacity.

We are sorry for having not discussed this point and we have added these discussions in our manuscript, please see the red-fond section on Page 5, 6, 19 in main text and Fig. S1, Table S5 and Note S6 in Supplementary Information

12. The Cl-I electrodes were prepared by electrochemically deposit I_2 on activated carbon matrix. Is this method feasible for preparing electrodes with large area.

Answer: Thank you for your valuable and helpful comments. The pouch cell was fabricated to demonstrate the scalability of the Cl-I electrode obtained by electrochemical deposition, which could be cycled to achieve the size of 80 cm^2 ($10 \text{ cm} * 8 \text{ cm}$, Fig. 5g). In addition, other applicable methods to deposit iodine into the host carbon can be developed, such as thermal-evaporation deposition. Thanks for your helpful suggestion.

We are sorry for having not discussed this point in our original manuscript, and according to your valuable comments, we have added more discussions on the necessity to develop more feasible electrode preparation methods for large-scale applications, please see the red-fond section on Page 20 and Note S7 in Supplementary Information.

13. Overall, it is suggested to submit this manuscript to specialized journals such as Chem, Angew. Chem. Or JMCA.

Answer: Thank you for your valuable and helpful comments. We have made a lot of efforts to address your comments and made corresponding changes. The key findings of our work are unprecedented, namely fixing the Cl^0/Cl^- reaction by I through chemical bonding to obtain the three-electron transfer electrode based on Cl-I two-redox centers. It is a new strategy to fix the Cl^-/Cl^0 reaction compared to the conventional one by applying a physical adsorption. In addition, the as-obtained performance of the non-metallic halogen electrode is superior compared to the general utilized cathode materials for Zn-based batteries, such as the MnO_2 , V_2O_5 and the Prussian blue analogs. Such new results for halogen-based electrodes with high capacity/energy density are of potentially crucial importance to the aqueous battery community. This report involves different aspects not only interhalogen chemistry but also the halogen materials, electrode/battery performance, and electrode development. Thus, we think it is suitable for the journal and we sincerely hope this current version could get your approval.

REVIEWERS' COMMENTS

Reviewer #3 (Remarks to the Author):

The authors have comprehensively revised the manuscript. I am satisfied with the revision. Meanwhile, I would like to suggest the authors to carefully polish the English language to correct all typo and grammar errors. This will help meet the high standard of Nature Communications.

Overall I recommend this manuscript to be accepted for publication in Nature Communication.

Dear Reviewers:

Thanks a lot for your constructive comments and suggestions concerning our manuscript entitled *“Development of rechargeable high-energy hybrid zinc-iodine aqueous batteries exploiting reversible chlorine-based redox reaction”*. Those comments are all valuable and very helpful for improving the quality of our paper. We have carefully studied your comments and substantially revised our manuscript which we hope to meet with approval.

We provide a point-by-point response below and the involved changes had been highlighted in the revised manuscript, where the original comments are shown in black and our responses are shown in blue.

=====

Reviewer #3:

The authors have comprehensively revised the manuscript. I am satisfied with the revision. Meanwhile, I would like to suggest the authors to carefully polish the English language to correct all typo and grammar errors. This will help meet the high standard of Nature Communications. Overall I recommend this manuscript to be accepted for publication in Nature Communication.

Answer: We are delighted on having addressed your concerns and thank you for the positive comments on our manuscript. Following your suggestion, we have polished our language.